# Transcriptome Analysis of Response to Zika Virus Infection in Two *Aedes albopictus* Strains with Different Vector Competence

**DOI:** 10.3390/ijms24054257

**Published:** 2023-02-21

**Authors:** Nan Jia, Yuting Jiang, Xianyi Jian, Tong Cai, Qing Liu, Yuan Liu, Dan Xing, Yande Dong, Xiaoxia Guo, Tongyan Zhao

**Affiliations:** Department of Vector Biology and Control, State Key Laboratory of Pathogen and Biosecurity, Beijing Institute of Microbiology and Epidemiology, Beijing 100071, China

**Keywords:** ZIKV, *Aedes albopictus*, vector competence, transcriptome analysis, cytochrome P450

## Abstract

Zika virus (ZIKV), which is mainly transmitted by *Aedes albopictus* in temperate zones, can causes serious neurological disorders. However, the molecular mechanisms that influence the vector competence of *Ae. albopictus* for ZIKV are poorly understood. In this study, the vector competence of *Ae. albopictus* mosquitoes from Jinghong (JH) and Guangzhou (GZ) Cities of China were evaluated, and transcripts in the midgut and salivary gland tissues were sequenced on 10 days post-infection. The results showed that both *Ae. albopictus* JH and GZ strains were susceptible to ZIKV, but the GZ strain was more competent. The categories and functions of differentially expressed genes (DEGs) in response to ZIKV infection were quite different between tissues and strains. Through a bioinformatics analysis, a total of 59 DEGs that may affect vector competence were screened—among which, cytochrome P450 304a1 (CYP304a1) was the only gene significantly downregulated in both tissues of two strains. However, CYP304a1 did not influence ZIKV infection and replication in *Ae. albopictus* under the conditions set in this study. Our results demonstrated that the different vector competence of *Ae. albopictus* for ZIKV may be determined by the transcripts in the midgut and salivary gland, which will contribute to understanding ZIKV–mosquito interactions and develop arbovirus disease prevention strategies.

## 1. Introduction

Zika virus (ZIKV), which belongs to the family *Flaviviridae*, genus *Flavivirus*, is a single-stranded RNA virus and primarily transmitted by *Aedes* mosquitos. ZIKV was discovered initially from a sentinel monkey in Uganda, Africa, in 1947 [1], and firstly detected in humans in 1952 [2]. In the past decade or so, ZIKV has continued to spread. It is currently recorded in 86 countries [3,4]. The clinical symptoms of Zika virus disease are variable, ranging from no or mild symptoms to severe neurological disorders such as microcephaly in infants born from infected mothers and Guillain-Barré syndrome in adults [5,6]. The spread of ZIKV poses a significant threat to public health. There is no specific drug or vaccine for ZIKV infection, and vector control remains the primary way to stop the spread of the virus [7,8]. However, with traditional vector control strategies becoming less effective, there is an urgent need for new methods to control the spread of arboviruses [9,10].

The female mosquito becomes infected with an arbovirus when it acquires a blood meal from an infected human. The ingested virus firstly invades the midgut tissue, where it replicates to produce viral particles. Then, the viral particles enter the hemolymph and spreads to secondary tissues, such as the trachea and salivary gland. Finally, the virus is released into salivary tubes and transmitted to uninfected vertebrate hosts by blood sucking for the next time [5,11,12]. Mosquito vector competence is influenced by the intrinsic factors and molecular mechanisms, and susceptibility to viruses varies among mosquito species and geographic strains [13,14,15]. This variability is driven by the compatibility of viruses with host factors and their ability to evade the action of the mosquito’s restriction factors, many of which are components of the insect’s innate immune system, such as Toll, immune deficiency (Imd), and the Janus kinase/signal transducer and activator of transcription (JAK/STAT) signaling pathways [5]. Activation of these pathways leads to alterations in transcription factors, resulting in the production of multiple antipathogen effector molecules [16,17]. Furthermore, the RNAi pathway, a key antiviral defense system, can degrade viral RNAs and plays an important role during arbovirus infections [18,19].

The molecular interactions between *Aedes aegypti* and ZIKV have been extensively studied, but the interactions between *Aedes albopictus* and ZIKV have largely not been elucidated. *Ae. albopictus* is an effective vector for ZIKV and has also been responsible for several outbreaks of important arboviruses such as dengue virus (DENV) and chikungunya virus (CHIKV) [20,21,22]. Yunnan and Guangdong Provinces are located in the southernmost part of China and are the main links between Southeast Asia and South America. With the increasing frequency of trade and people exchanges in recent years, the epidemic situation cannot be ignored. Therefore, in this study, *Ae. albopictus* mosquitoes from Jinghong City (JH), Yunnan Province and Guangzhou City (GZ), Guangdong Province were collected, and their vector competence for ZIKV was assessed. The RNA-seq technique was used to analyze the changes in the transcriptome of the midgut and salivary gland tissues of the two strains of *Ae. albopictus* after infection with ZIKV. Furthermore, our comparative analysis of the ZIKV infection-responsive transcriptomes identified potential ZIKV infection-responsive genes, which could contribute to the development of new insect-borne disease prevention strategies.

## 2. Results

### 2.1. Vector Competence of Ae. albopictus JH and GZ Strains for ZIKV

To evaluate the vector competence of *Ae. albopictus* JH and GZ strains for ZIKV, a low dose of 1 × 10^6^ PFU/mL and a high dose of 1 × 10^7^ PFU/mL ZIKV were used to orally infect the mosquitoes. Viral RNA in the midgut and salivary gland was detected at 4, 7, 10 and 14 days post-infection (dpi), and the infection rate and viral RNA copies were assessed (Figure 1).

When the mosquitoes were infected with a low dose of ZIKV, viral RNA was detected as positive in all tissues at all sampling days, except the salivary gland of the JH strain at 4 and 7 dpi. The Infection rate in the midgut of the JH strain was maintained at a relatively low level, ranging from 30.0~33.3% at 4~10 dpi, and suddenly reached 80.0% at 14 dpi, but it was maintained at a high level of 80.0% to 96.7% at all sampling days in the midgut of the GZ strain (Figure 1A). The infection rate in the salivary gland of the GZ strain gradually increased from 3.3% at 4 dpi to 46.7% at 14 dpi. Viral RNA could not be detected until 10 dpi in the salivary gland of the JH strain with a 3.3% infection rate, and it reached 6.7% at 14 dpi (Figure 1B). The infection rate was significantly higher in the midgut of the GZ strain at 4, 7 and 10 dpi and in the salivary gland at 14 dpi when compared to that of the JH strain. The viral RNA copies in the positive samples were calculated and compared, but there was no significant difference between the JH and GZ strains in both the midgut and salivary gland (Figure 1C,D).

When the mosquitoes were infected with a high dose of ZIKV, the infection rate in the midgut of the JH strain gradually increased from 50.0% at 4 dpi to 83.3% at 14 dpi, but it was maintained at a high level of 83.3% to 96.7% in the GZ strain at all sampling days (Figure 1E), similar to the low-dose infection experiments. The infection rate in the salivary gland exhibited a pattern of progressive increase both in the JH and GZ strain. It increased from 0.0% at 4 dpi to 26.7% at 14 dpi in the JH strain and from 10.0% at 4 dpi to 50.0% at 14 dpi in the GZ strain (Figure 1F). The infection rate of the GZ strain was significantly higher in the midgut at 4 dpi and in the salivary gland at 10 dpi than that of the JH strain. There was a significant difference in the viral RNA copies in the midgut tissues, with a higher load in the GZ strain than the JH strain (Figure 1G). However, it was not significantly different between the two strains in the salivary gland (Figure 1H).

Taken together, it was shown that the *Ae. albopictus* GZ strain was more competent for ZIKV than the JH strain.

### 2.2. mRNA Sequencing of Ae. albopictus JH and GZ Strains Infected with ZIKV

Albeit both the *Ae. albopictus* JH and GZ strain were susceptible to ZIKV infection, the GZ strain was more competent for ZIKV than the JH strain. To screen genes that potentially influence the vector competence of *Ae. Albopictus* for ZIKV, the transcription profile in the midgut and salivary gland of these two mosquito strains that were infected with ZIKV (1 × 10^7^ PFU/mL) at 10 dpi were obtained by Illumina sequencing, and the mosquitoes that were fed with uninfected blood meal were used as the control. A total of eight RNA-seq libraries were created, and 23.92 M raw reads were obtained from each library. Then, some reads were discarded in different libraries due to their low-quality scores or lack of adapter sequences. Finally, about 23.82–23.83 M clean reads were maintained in each library.

### 2.3. Differentially Expressed Genes (DEGs) in Response to ZIKV Infection in Two Ae. albopictus Strains and Function Analysis

Transcripts in the different libraries were analyzed, and fold changes of genes expression in the ZIKV-infected tissues compared to the non-infected group were calculated. The transcripts with a false discovery rate (FDR) < 0.001 in the group comparisons were defined as significantly regulated and those with FDR < 0.001 and |log2FC| > 1 as DEGs. A total of 211, 297, 228 and 1287 DEGs were identified in the midgut and salivary gland of *Ae. albopictus* JH and GZ strains in response to ZIKV infection, respectively (Figure 2). Interestingly, the most alterations in the mRNA profile were found in the salivary gland of the GZ strain, and the changes in mRNA expression were predominately downregulated (Figure 2D). To further analyze the related functions of DEGs and identify the biological pathways that play a key role in the biological processes, with the aim of revealing and understanding the basic molecular mechanism, Go and KEGG pathway analyses of the DEGs were performed.

The GO functions enriched by DEGs were largely consistent in both two tissues and two strains (Appendix A). The DEGs were mostly enriched in (1) the biological process (BP) terms: cell process, metabolic process, biological regulation, regulation of biological processes, multicellular organismal process and response to stimulus; (2) the cellular component (CC) terms: cell, cell part, membrane, membrane part and organelle and (3) the molecular function (MF) terms: binding, catalytic activity and transporter activity.

Unlike the GO analysis, the KEGG enrichment results showed that the KEGG pathways enriched were different between two tissues and two strains (Appendix A). For the JH strain, DEGs in the midgut are mainly involved in proximal tubule bicarbonate reclamation, tyrosine metabolism, dorsoventral axis formation and the glucagon signaling pathway (Appendix A). DEGs in the salivary gland are mainly involved in phototransduction—fly, the oxytocin signaling pathway, metabolism of xenobiotics by cytochrome P450 and drug metabolism—and other enzymes (Appendix A). For the GZ strain, DEGs in the midgut are mainly involved in galactose metabolism, pentose and glucuronate interconversions, the Fanconi anemia pathway and salivary secretion (Appendix A). DEGs in the salivary gland are mainly involved in pancreatic secretion, protein digestion and absorption, the metabolism of xenobiotics by cytochrome P450, adrenergic signaling in cardiomyocytes, etc. (Appendix A).

### 2.4. Comparison of DEGs in Different Tissues and Strains

The midgut and salivary gland play important roles in arbovirus infection and transmission via mosquitoes. Viruses have to overcome infection barrier and escape barrier in these two tissues to be secreted in the saliva and ready for infection of the vertebrate by blood sucking [23]. However, it was found that DEGs after ZIKV infection were quite different between the midgut and salivary gland in both the JH and GZ strains. There are only 7 (1.4%) DEGs shared by two tissues in the JH strain and 53 (3.6%) DEGs in the GZ strain. For these consensus DEGs, only one DEG was found in both the JH and GZ strains, namely LOC109405426 (cytochrome P450 304a1, CYP304a1) (Appendix A). The function of those 59 genes is summarized in Table 1. Differences in the DEGs between the JH and GZ strains in the midgut or salivary gland are also compared. There are only 24 (5.8%) consensus DEGs shared by two strains in the midgut and 41 (2.7%) consensus DEGs in the salivary gland (Appendix A). Similar to the KEGG enrichment analysis, these results showed that the transcriptome expression in response to virus infection were quite different between tissues and among mosquito strains.

### 2.5. RT-qPCR Validation

To validate the results of deep sequencing, 10 genes, including LOC109405426, in Table 1 were selected for RT-qPCR verification. The results showed that the expression levels of the DEGs obtained by RNA-Seq and RT-qPCR were consistent, indicating the results from the RNA-Seq were reliable (Figure 3).

### 2.6. CYP304a1 Did Not Influence ZIKV Infection and Replication in Ae. albopictus

As previously described, only LOC109405426 (CYP304a1) was significantly downregulated in both two tissues and two strains after ZIKV infection, so it was chosen for further examination. The gene was knocked down by siRNA via thoracic microinjection, and its relative expression was examined 3 days after injection. The results showed that the transcription of CYP304a1 was significantly decreased after siRNA inoculation compared to the GFP control group (Figure 4A). Three days after gene silencing, the ZIKV suspension was microinjected into the mosquitoes, and the viral loads were assessed on 1 and 3 dpi via RT-qPCR. The results showed that the ZIKV load on 3 dpi was significantly higher than that on 1 dpi (*p* < 0.0001) in both the JH and GZ strains, but there was no significant difference between the CYP304a1 interference group and GFP group (Figure 4B,C), indicating that CYP304a1 did not influence ZIKV infection and replication in *Ae. albopictus*.

## 3. Discussion

Jinghong and Guangzhou, which are important port cities in South China, have close trade relations with Southeast Asia and South America and are the risk sites for the introduction of Zika virus. *Ae. albopictus* is a common mosquito species in tropical and subtropical areas of China and is an important vector for DENV, ZIKV and other Flaviruses. This study evaluated the vector competence of the *Ae. albopictus* Jinghong strain and Guangzhou strain for ZIKV, and the results showed that the GZ strain was a highly efficient vector for ZIKV, consistent with a previous study [15]. However, the JH strain showed a relative lower vector competence. When it was infected with a low dose of ZIKV, the infection rate of the midgut in the JH strain was significantly lower than that of GZ, and viral RNA was detected in the salivary gland until 10 dpi. When infected with a high dose of the virus, the viral RNA copies in the midgut of the JH strain were significantly lower than that of GZ, and viral RNA was detected in the salivary gland until 7 dpi. These results indicated that the JH strain has stronger resistance to ZIKV infection. However, in the high-dose experiment, there was no statistical difference of the virus infection rate and copy number in the salivary glands between two mosquito strains at the late infection stage (14 dpi), so the risk of ZIKV transmission by *Ae. albopictus* in Jinghong City should not be ignored.

Different vector competence for ZIKV between geographic strains of the same mosquito species have been previously reported. For example, *Ae. aegypti* from Brazil, the Dominican Republic, and the United States were fed with artificial blood meals containing ZIKV, but only mosquitoes from the Dominican Republic transmitted the ZIKV Cambodia and Mexica strains [24]. Field *Ae. aegypti* from three Pacific islands were collected and orally exposed to ZIKV, and the results showed that the ZIKV infection rate was heterogeneous between the populations [25]. The vector competence of nine *Ae. albopictus* populations in China for DENV-2 was evaluated, and it was shown that significant differences of viral RNA copies existed among different populations [26]. The factors that influence vector competence are complicated, and the genetic background is one of them. In order to find the potential genes related to vector competence, a transcriptome sequencing was performed, and the DEGs in the midgut and salivary glands of two *Ae. albopictus* strains after ZIKV infection were screened and compared.

The results showed that the DEGs were very different between two tissues and two mosquito strains. When comparing DEGs between tissues, the consensus DEGs account for only 1.4% of the total in the two tissues of the JH strain and 3.6% in the GZ strain. When comparing DEGs between strains, there were only 5.8% consensus DEGs shared by two strains in the midgut and 2.7% in the salivary gland. In the GO and KEGG pathway analyses, DEGs screened in four libraries were very similar in composition (Appendix A) but differed greatly in function (Appendix A). Subsequently, DEGs with the same regulated direction in both the midgut and salivary gland were screened, because genes that were upregulated in one tissue but downregulated in the other would make the following analysis and gene silencing validation complicated and contradictory. Such an analysis strategy might have omitted genes that influence the vector competence. However, the genes screened by this way had the highest probability of being related to vector competence. Finally, a total of 59 DEGs were screened, including defensin-A, RNA polymerase II subunit Rpb1, cytochrome P450, etc.

Defensin-A, which was significantly downregulated after ZIKV infection both in the midgut and salivary gland of the GZ strain, is a member of the defensin family and the first biological line of defense against pathogen invasion [27]. Studies have shown that mosquito defensins are primarily active against Gram-positive bacteria [28]. However, recent studies have found that the defensin gene family plays a role after the mosquito is infected by the virus. For example, defensin-A is significantly reduced in DENV-1-infected *Ae. aegypti*, and it has been speculated that DENV-1 may inhibit the expression of certain factors required to induce defensin mRNA expression through the Toll pathway or directly target and inhibit gene transcription [29]. This is similar to the results of the present study and suggests that the defensin family is also involved in the defense process against ZIKV infection in *Ae. albopictus*. However, how ZIKV inhibit its expression and what effector molecules are involved remain to be determined.

In addition, RNA polymerase II subunit Rpb1 was significantly downregulated in both the midgut and salivary gland of the GZ strain after ZIKV infection. At present, little is known about the function of this protein during infection. However, studies have reported that, after Semliki Forest Virus, Sindbis Virus or CHIKV infection, mRNA transcription in cells was inhibited by rapid degradation of the Rpb1 catalytic subunit of RNA polymerase II, thereby inhibiting cell antiviral reaction [30]. The results of this study indicated that RNA polymerase II subunit Rpb1 was involved in the process of ZIKV infection in *Ae. Albopictus*, but its clear mechanism and interaction with other molecules require further investigation.

Among the 59 genes screened, one was significantly downregulated in two strains and two tissues, namely CYP304a1. The CYP enzymes are membrane-bound hemoproteins that play a pivotal role in the detoxification of xenobiotics, cellular metabolism and homeostasis [31]. It was reported that this gene played pivotal roles in the tolerance to toxic leaf litter for *Ae. aegypti* larvae [32], detoxification of pyrethroid insecticides for *Anopheles minimus* [33] and resistance to permethrin for *Culex quinquefasciatus* [34]. However, the relationship between mosquito CYP and virus infection has not been reported previously. In order to verify the effect of CYP304a1 on mosquito vector competence, interfering RNA for this gene were designed and intrathoracically injected into the mosquito to knock down its expression. Three days after interference, mosquitoes were infected with ZIKV by intrathoracic injection, and viral RNA were detected 1 and 3 days after infection. The results showed that the copy number of ZIKV virus was higher at 3 dpi than 1 dpi (*p* < 0.0001), but no significant difference between the CYP304a1 interference group and GFP group was found in this study, indicating that CYP304a1 had no effect on virus infection and replication under the conditions set in this study. However, the virus infection route and dose might have effects on the results. For example, virus infection by intrathoracic injection bypassed the midgut infection and escape barrier, where CYP304a1 may play a role. The high dose of virus injection used in this study (3000 PFU/mosquito) may have overloaded the antiviral effect of CYP304a1 or covered up its effect of promoting virus infection. Other infection routes (e.g., oral infection) or a lower virus dose may be used in future studies to further confirm the role of CYP304a1.

## 4. Materials and Methods

### 4.1. Mosquito Strain

The *Aedes albopictus* Jinghong strain was originally collected from Jinghong City, Yunnan Province (GPS location: 21°26′ N and 100°25′ E), in 2019. The *Ae. albopictus* Guangzhou strain was originally collected from Guangzhou City, Guangdong Province (GPS location: 23°07′ N and 113°16′ E), in 2019. Both two mosquito strains were reared under standard insectary conditions at 26 ± 1 °C and 75 ± 5% relative humidity, with a photoperiod of 14 h light:10 h dark cycles. Prior to the infectious feed, adult mosquitoes were provided with 8% sucrose solution.

### 4.2. Cell Line and Virus Stocks

C6/36 (*Ae. albopictus*) cells were maintained in our laboratory and were cultured in RPMI 1640 medium (Gibco, Shanghai, China) supplemented with 10% fetal bovine serum (FBS) (Gibco, Shanghai, China) and 1% penicillin/streptomycin (Gibco, Shanghai, China) at 28 °C in an incubator of 5% CO_2_.

The ZIKV SZ01 strain used in this study was obtained from the Microbial Culture Collection Center of the Beijing Institute of Microbiology and Epidemiology. This virus was originally isolated from a patient who returned from Samoa to China in 2016 (GenBank accession number: KU866423) [35]. The virus has been passaged in the C6/36 cell lines six times.

### 4.3. Oral Infection of Mosquitoes

Virus-infected blood meals were prepared by mixing 1:1 mouse blood and ZIKV SZ01 strain suspension supplemented with 2% FBS and 1% heparin sodium. Seven-day-old adult female mosquitoes that had been starved for 18 h were fed with this infected blood meal using a Hemotek membrane feeding system. The blood meal was kept at 37 °C during feeding. The non-infected group was supplied with a 1:1 mixture of mouse blood and RPMI 1640 medium supplemented with 2% FBS and 1% heparin sodium. After 1 h of feeding, mosquitoes were cold-anesthetized, and blood-engorged mosquitoes were transferred to and maintained in the standard rearing conditions.

### 4.4. Mosquitoes Processing and RNA Extraction

Thirty blood-engorged mosquitoes were respectively sampled at 4, 7, 10 and 14 dpi. The midguts and salivary glands of mosquitoes were dissected and collected carefully with sterile dissecting needles and individually transferred into 1.5 mL microtubes containing 1 mL of RNAiso Plus (TaKaRa, Dalian, China). Then, the total RNA from the midguts and salivary glands was extracted according to the manufacturer’s instructions.

### 4.5. ZIKV Detection

ZIKV genomic RNA was detected using the GoTaq^®^ Probe 1-Step RT-qPCR System (Promega, Dalian, China), including forward primer: 5′-AAGTTTGCATGCTCCAAGAAAAT-3′, reverse primer: 5′-CAGCATTATCCGGTACTCCAGAT-3′ and probe: 5′-FAM-ACCGGGAAGAGCATCCAGCCAGA-TAMRA-3′ [36]. The following reagents were used for the RT-qPCR reactions: 2 μL of RNA sample, 10 μL of GoTaq^®^ Probe qPCR Master Mix, 0.4 μL of GoScript™ RT Mix, 1 μL forward primer, 1 μL reverse primer, 1 μL probe and 4.6 μL of nuclease-free water to yield a 20 μL final reaction volume. Amplification reactions were performed in the QuantStudio™ 7 Flex Real-Time PCR system (Thermo Fisher Scientific, Waltham, MA, USA) and programmed as follows: 1 cycle at 45 °C for 15 min, 95 °C for 10 min, 40 cycles at 95 °C for 15 s and 60 °C for 1 min. Virus RNA copies were calculated by generating a standard curve using a recombinant plasmid-containing virus segment insertion.

### 4.6. Sample Preparation for mRNA-seq

Seven-day-old female *Ae. albopictus* JH and GZ strains were fed with a ZIKV-infected blood meal or a blood meal devoid of ZIKV, as described previously. The midgut and salivary gland were dissected at 10 dpi. A total of 8 groups (2 strains × 2 tissues × 2 kinds of blood meal) were included for mRNA seq library preparation. Each group contained approximately 100 mosquitoes, from which tissues were collected into a 1.5 mL RNase-free microcentrifuge tube containing 1 mL RNAiso Plus (TaKaRa, Dalian, China) and stored at −80 °C until the subsequent RNA extraction.

### 4.7. RNA Extraction, mRNA Library Preparation and Sequencing

The RNA extraction, library preparation and sequencing analyses were performed by the BGI Company (Shenzhen, China). The total RNA was extracted from 8 groups using RNAiso Plus according to the manufacturer’s protocols. The quality and quantity of RNA were measured by the Agilent 2100 Bioanalyzer System (Agilent Technologies, Inc., Santa Clara, CA, USA). Each RNA sample was divided into two parts, with one used for mRNA library preparation and sequencing and the second part used for RT-qPCR validation. Oligo (dT) magnetic beads were used for the enrichment of mRNAs with a poly-A tail. Purified mRNA was fragmented into small pieces with fragment buffer at the appropriate temperature. Then, first-strand cDNA was generated using random hexamer-primed reverse transcription, followed by a second-strand cDNA synthesis. The purified double-stranded cDNA was repaired, A-tails were added to the ends and the products were purified again after PCR amplification to finally obtain a single-stranded circular DNA library. The quality of cDNA was checked using the Agilent 2100 Bioanalyzer system. mRNA sequencing was performed using the Illumina genomic analyzer.

### 4.8. Bioinformatics

To ensure the quality and reliability of the data analysis, it was necessary to filter the original data. Reads with adapters, undetermined base information and low quality were removed with SOAPnuke v1.5.2 [37]. Clean reads were mapped to the reference genome using HISAT2 v2.0.4 [38] to obtain the localization information of the reads on the reference genome. Bowtie2 (v2.2.5) [39] was applied to align the clean reads to the reference coding gene set; then, the expression level of the gene was calculated by RSEM (v1.2.12) [40]. Finally, GO (http://www.geneontology.org/, accessed on 11 May 2022) and KEGG (https://www.kegg.jp/, accessed on 11 May 2022) enrichment analyses of the annotated DEGs were performed by Phyper (https://en.wikipedia.org/wiki/Hypergeometric_distribution, accessed on 11 May 2022) based on the Hypergeometric test.

### 4.9. Expression Profiling of mRNAs in Response to ZIKV

An algorithm was used to identify differentially expressed mRNAs between ZIKV-infected samples and non-infected samples. p(χ) = e^−λ^λ^χ^/χ!, where χ is defined as the number of reads from mRNA and λ is the real transcripts of the mRNA. The method was described by Audic et al. [41]. When the FDR was <0.001, changes in the mRNA expression were considered to be significant. mRNAs with log2FC > 1 were designated as being significantly upregulated, and mRNAs with log2FC ≤ 1 were designated as significantly downregulated.

### 4.10. RT-qPCR Validation of RNA-Seq Data

To confirm the RNA-Seq data, the expression of 10 mRNA transcripts was verified by 2-step RT-qPCR. Firstly, RNA was reverse-transcribed into cDNA with the PrimeScript™ RT Reagent Kit with gDNA Eraser (TaKaRa, Dalian, China). One microliter of RNA template, two microliters of 5× gDNA Eraser Buffer, one microliter of gDNA Eraser and six microliters of RNase-free water were mixed and incubated at 42 °C for 2 min. Then, 1 µL of PrimeScript RT Enzyme Mix I, 1 µL of RT Primer Mix, 4 µL of 5× PrimeScript Buffer II and 4 µL of RNase-free water were added to the above reaction solution, and the mixture was incubated at 37 °C for 15 min, then 85 °C for 5 s. The products were used as templates for qPCR validation in the next step. qPCR was performed using the PerfectStart™ Green qPCR SuperMix Kit (Transgene, Beijing, China). The reaction system consisted of 2 µL of cDNA template, 10 µL of SuperMix, 0.4 µL of Passive Reference Dye II, 0.4 µL F/R primer and 6.8 µL of RNase-free water. qPCR was performed using the QuantStudio™ 7 Flex Real-Time PCR system. The reaction procedure was as follows: 94 °C for 30 s for 1 cycle; 94 °C for 5 s and 60 °C for 30 s for 40 cycles for the dissociation stage. The 2^−∆∆CT^ method [42] was used to analyze the qPCR results. The primer sequences of selected mRNA transcripts and the reference gene actin are shown in Appendix A.

### 4.11. Gene Silencing and Viral Infection by Thoracic Microinjection

CYP304a1 and GFP siRNA were designed by the DSIR website (http://biodev.extra.cea.fr/DSIR/DSIR.html, accessed on 9 September 2022), and the sequences are shown in Appendix A. For gene silencing, female *Ae. albopictus* mosquitoes were cold-anaesthetized on a cold tray, and 0.01 nmol/300 nL of siRNA were injected into their thoraxes. The injected mosquitoes were allowed to recover 3 days under standard rearing conditions for viral infection. Then, the mosquitoes were thoracically microinjected with 300 nL of 1 × 10^7^ PFU/mL ZIKV suspension and were then maintained in rearing conditions. The total RNA of the mosquitoes was extracted 3 days after siRNA injection for gene silencing validation or 1 and 3 days after virus infection for ZIKV detection. The methods of RNA extraction, CYP304a1 and ZIKV detection were the same as previously described in this study. The sample sizes of siRNA microinjection, ZIKV infection and ZIKV detection are summarized in Appendix A.

### 4.12. Statistical Analysis

The data analysis was performed using GraphPad Prism (Version 8.0, GraphPad Software, San Diego, CA, USA). In the vector competence evaluation experiment, the infection rate between two strains was compared by Fisher’s exact test. Viral RNA copies in two *Ae. albopictus* strains were compared with the non-parametric Mann–Whitney test. In the CYP304a1 silencing experiment, the relative expression of CYP304a1 between two groups was compared by the non-parametric Mann–Whitney test. The viral loads in different strains at different times post-infection were analyzed by 2-way ANOVA with Šídák’s multiple comparisons test. The normality and heteroscedasticity of the residuals were evaluated by the Shapiro–Wilk test and Spearman’s test, respectively. *p*-values lower than 0.05 were considered statistically significant.

## 5. Conclusions

In this study, the vector competence of *Ae. albopictus* from Jinghong and Guangzhou Cities were evaluated, and the results showed that both strains were susceptible to ZIKV, but the GZ strain was more competent. The transcription profile in the midgut and salivary gland of these two mosquito strains in response to ZIKV infection was examined and showed significant differences. A total of 59 DEGs that were simultaneously up- or downregulated in the midgut and salivary gland were screened out and were thought to influence the vector competence of *Ae. albopictus*. Particularly, CYP304a1 was significantly downregulated in both tissues and strains and might play a role in ZIKV–mosquito interactions, although it did not influence the virus infection and replication under the conditions set in this study. Our research provided new insights into mosquito–virus interactions, which could contribute to new strategy developments for arbovirus disease prevention.

## Figures and Tables

**Figure 1 ijms-24-04257-f001:**
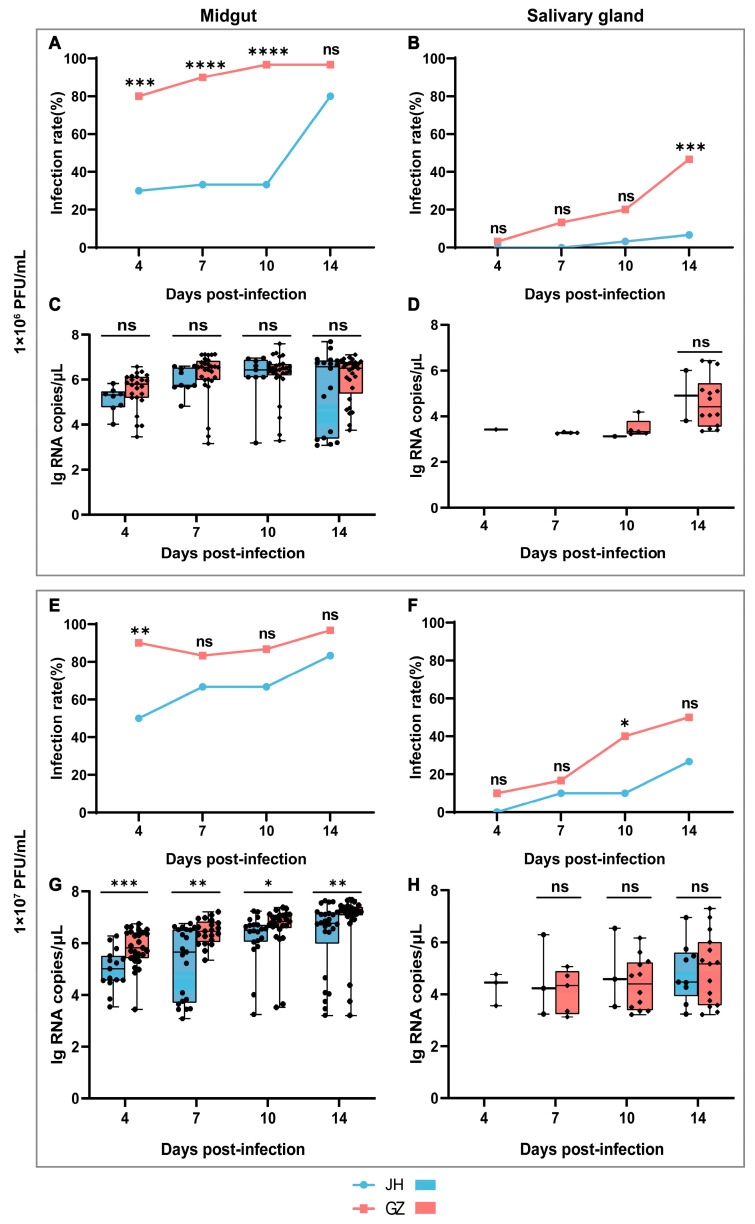
Vector competence of *Ae. albopictus* JH and GZ strains to ZIKV. The upper panel shows the results of the infection rate and viral RNA copies in two *Ae. albopictus* strains infected with a low dose of ZIKV (1 × 10^6^ PFU/mL), and the lower panel shows the results of a high dose of ZIKV infection (1 × 10^7^ PFU/mL). Infection rate in the midgut (**A**,**E**) and salivary gland (**B**,**F**) of two mosquito strains. Viral RNA copies (log 10 transformation) in the midgut (**C**,**G**) and salivary gland (**D**,**H**) of two mosquito strains. The infection rate of the two *Ae. albopictus* strains were compared using Fisher’s exact test, and viral RNA copies were compared using the non-parametric Mann–Whitney test (Mann–Whitney test requires at least two values in each group; the data that did not meet this standard were not compared) to determine the differences between the two strains at different times, since the viral RNA copies did not conform to a normal distribution (Shapiro–Wilk test). Box plots show the median and the 25th to 75th percentiles, and the whiskers denote the minimum and maximum values. * *p* < 0.05, ** *p* < 0.01, *** *p* < 0.001 and **** *p* < 0.0001, ns: no significant difference.

**Figure 2 ijms-24-04257-f002:**
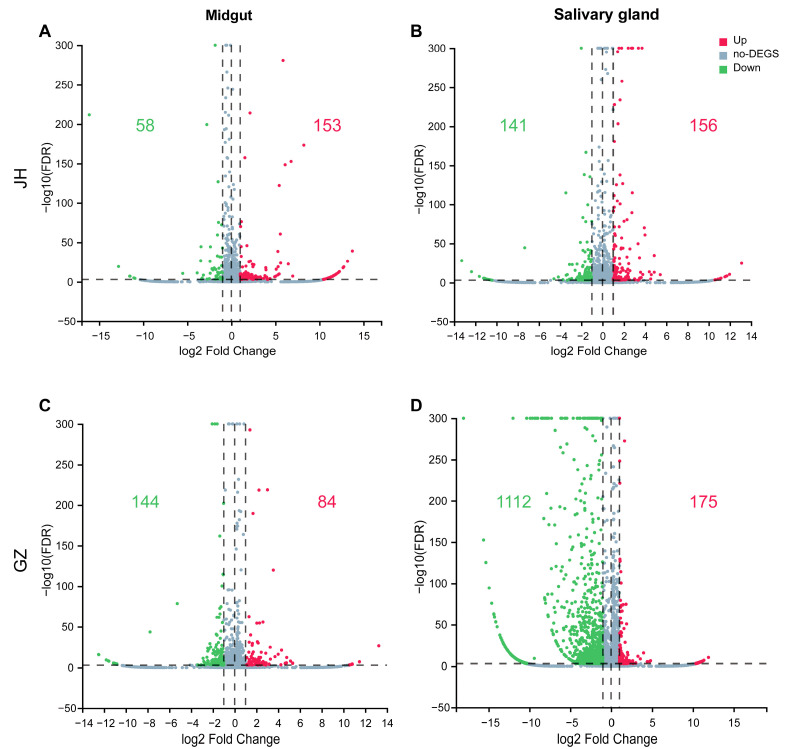
Volcano plot of DEGs (|log2FC| > 1, FDR < 0.001). mRNA expression levels in the midgut (**A**,**C**) and salivary gland (**B**,**D**) of *Ae. albopictus* JH and GZ strains infected with ZIKV were compared to those of the non-infected. Red and green plots in-dicate significantly upregulated and downregulated genes, respectively. Light blue plots indicate genes that were not significantly different between the infected and non-infected groups. Number of up- or downregulated genes was shown in the relative quadrant with the same color.

**Figure 3 ijms-24-04257-f003:**
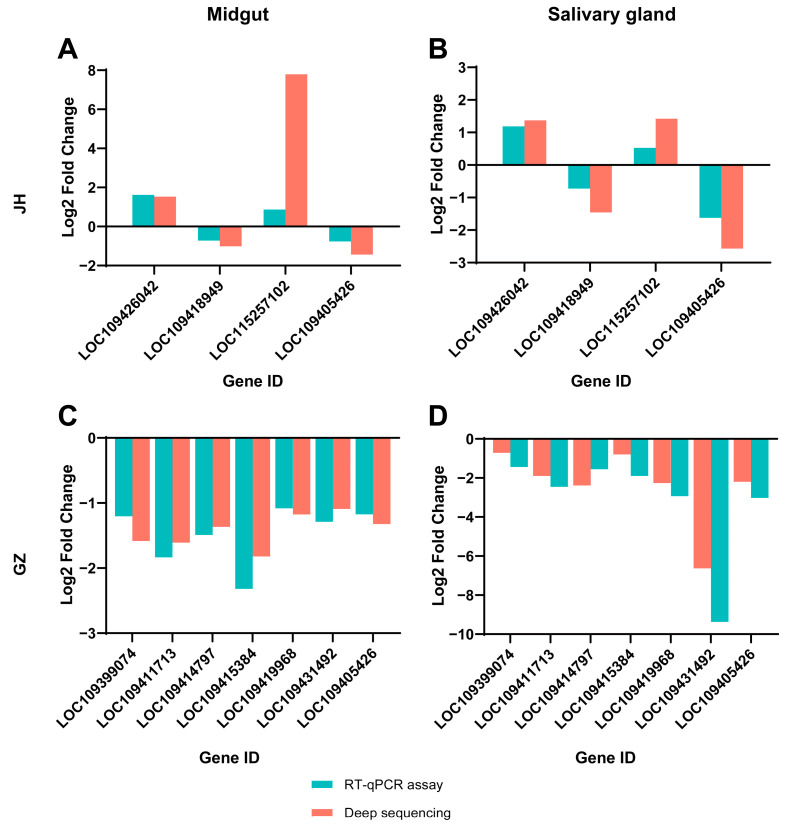
RT-qPCR and transcriptome analysis of 10 DEGs. Four DEGs in the JH strain (**A**,**B**) and seven DEGs in the GZ strain (**C**,**D**) (including one gene that was differentially expressed in both strains) were selected for RT-qPCR validation. The fold changes determined from the RT-qPCR assay and deep sequencing are shown.

**Figure 4 ijms-24-04257-f004:**
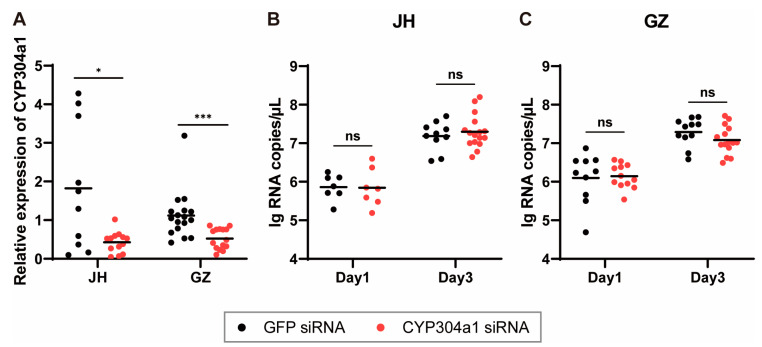
CYP304a1 silencing and ZIKV infection in *Ae. albopictus*. (**A**) Abundance of CYP304a1 after RNA interference. Mosquitoes were microinjected with GFP and CYP304a1 siRNA, and the relative expression of CYP304a1 in the mosquitoes was detected by RT-qPCR. One dot represents 1 mosquito, and the horizontal line represents the median of the results. Differences between two groups were compared by the non-parametric Mann–Whitney test. The siRNA knockdown efficiency of CYP3304 a1 in the JH and GZ strains was 65.5% and 53.5%, respectively. ZIKV RNA copies were in the *Ae. albopictus* JH (**B**) and GZ (**C**) strains. The ZIKV suspension was inoculated by thoracic microinjection at 3 days post-CYP304a1 silencing. The viral load was assessed at 1 and 3 dpi through RT-qPCR. One dot represents 1 mosquito, and the horizontal line represents the mean of the results. The data were analyzed by 2-way ANOVA with Šídák’s multiple comparisons test. * *p* < 0.05 and *** *p* < 0.001. ns, no significant difference.

**Table 1 ijms-24-04257-t001:** DEGs that are regulated in the same direction in both the midgut and salivary gland tissues of the JH and GZ strains after ZIKV infection.

Gene ID	Strain	Midgut		Salivary Gland		Gene Description
		log2FC	lg FDR	log2FC	lg FDR	
LOC109426042	JH	1.53	−157.38	1.37	−104.46	peptide methionine sulfoxide reductase
LOC115257102	JH	11.15	−5.30	1.42	−5.28	uncharacterized LOC115257102
LOC115263986	JH	1.05	−70.14	1.11	−46.59	selenide, water dikinase-like
LOC109413251	JH	−1.50	−9.38	−1.69	−6.57	solute carrier family 25 member 35-like
LOC109418949	JH	−1.02	−42.82	−1.46	−51.82	ecdysone-induced protein 74 EF-like
LOC109422129	JH	−12.81	−19.36	−12.36	−13.92	F-box/LRR-repeat protein 4
LOC109403309	GZ	2.13	−30.39	1.80	−74.66	Na(+)/H(+) exchange regulatory cofactor NHE-RF2
LOC109406052	GZ	1.34	−30.96	2.03	−13.35	V-type proton ATPase subunit C-like
LOC109408676	GZ	13.24	−26.59	11.91	−10.22	multidrug resistance protein homolog 49-like
LOC109417532	GZ	1.68	−189.58	1.41	−11.67	lipopolysaccharide-induced tumor necrosis factor-alpha factor homolog
LOC115264596	GZ	5.33	−5.18	10.55	−3.67	Pre-mRNA-splicing factor CWC22 homolog
LOC115267308	GZ	1.88	−7.17	1.91	−3.73	homeobox protein 5-like
LOC109399074	GZ	−1.58	−6.96	−1.43	−4.85	Defensin-A
LOC109399245	GZ	−1.64	−3.93	−9.11	−∞	maternal protein exuperantia-2-like
LOC109401933	GZ	−1.52	−7.23	−2.13	−3.92	sodium-independent sulfate anion transporter-like
LOC109402161	GZ	−1.54	−4.88	−2.11	−3.71	adhesion G-protein coupled receptor G2
LOC109402605	GZ	−1.13	−55.16	−4.17	−184.28	uncharacterized LOC109402605
LOC109402737	GZ	−1.41	−6.73	−2.13	−16.93	Proline-rich extension-like protein EPR1
LOC109405806	GZ	−1.00	−7.11	−3.49	−65.82	fatty acyl-CoA reductase wat
LOC109406984	GZ	−1.86	−5.98	−3.58	−4.51	probable 4coumarate--CoA ligase 3
LOC109407962	GZ	−2.79	−3.38	−10.52	−3.51	myosin light chain kinase 2, skeletal/cardiac muscle-like
LOC109408573	GZ	−2.07	−∞	−10.33	−∞	protein FAM133A-like
LOC109408574	GZ	−1.32	−18.95	−12.73	−19.52	Sialidase-like
LOC109409947	GZ	−2.55	−12.31	−5.78	−43.89	cell wall protein DAN4-like
LOC109410284	GZ	−1.16	−100.29	−2.13	−68.19	UDP-glucuronosyltransferase 2B18-like
LOC109410780	GZ	−1.69	−40.73	−6.74	−120.48	uncharacterized LOC109410780
LOC109410802	GZ	−1.98	−9.21	−5.75	−13.88	A-agglutinin anchorage subunit-like
LOC109411630	GZ	−1.77	−15.02	−1.51	−13.05	uncharacterized LOC109411630
LOC109411671	GZ	−1.32	−3.94	−2.23	−6.05	Neuroligin-4, Y-linked-like
LOC109411713	GZ	−1.61	−∞	−2.45	−88.92	uncharacterized LOC109411713
LOC109413271	GZ	−1.58	−18.33	−2.25	−18.04	delta(9)-fatty-acid desaturase fat-7-like
LOC109414797	GZ	−1.37	−161.71	−1.56	−18.91	sodium/potassium/calcium exchanger 3-like
LOC109415384	GZ	−1.82	−∞	−1.90	−3.98	flocculation protein FLO11-like
LOC109417111	GZ	−10.99	−4.71	−11.52	−7.87	deoxynucleoside kinase
LOC109418600	GZ	−1.04	−114.05	−1.61	−∞	uncharacterized LOC109418600
LOC109418601	GZ	−1.21	−4.40	−2.66	−7.50	uncharacterized LOC109418601
LOC109419596	GZ	−1.12	−3.93	−1.73	−8.32	ABC transporter G family member 20
LOC109419968	GZ	−1.18	−27.22	−2.93	−242.82	DNA-directed RNA polymerase II subunit rpb1-like
LOC109420639	GZ	−1.26	−17.77	−1.95	−3.33	zinc transporter 2-like
LOC109421462	GZ	−1.35	−65.71	−1.40	−248.73	ejaculatory bulb-specific protein 3-like
LOC109425972	GZ	−1.74	−24.71	−13.97	−47.35	cytochrome P4506a2-like
LOC109428862	GZ	−2.37	−22.36	−2.25	−7.75	glutamic acid-rich protein-like
LOC109429835	GZ	−1.03	−7.16	−2.04	−3.51	uncharacterized LOC109429835
LOC109430556	GZ	−1.49	−30.55	−2.17	−3.07	solute carrier organic anion transporter family member 74D
LOC109430578	GZ	−1.30	−4.92	−7.08	−37.96	bifunctional endo-1,4-beta-xylanase XylA-like
LOC109430774	GZ	−2.21	−25.08	−5.17	−64.21	UDP-glucuronosyltransferase 1-6-like
LOC109430836	GZ	−2.47	−19.93	−2.66	−11.08	glutaminase kidney isoform, mitochondrial-like
LOC109431492	GZ	−1.07	−114.64	−14.94	−94.39	flocculation protein FLO11-like
LOC115253999	GZ	−11.62	−7.86	−12.30	−14.11	protein LSM14 homolog B-like
LOC115255052	GZ	−1.13	−10.06	−11.40	−7.28	sodium-coupled monocarboxylate transporter 2-like
LOC115255085	GZ	−2.27	−29.17	−11.35	−6.98	zinc transporter 2-like
LOC115256378	GZ	−1.08	−3.33	−10.97	−5.24	uncharacterized LOC115256378
LOC115262105	GZ	−2.00	−9.41	−1.77	−18.39	circadian clock-controlled protein-like
LOC115262616	GZ	−1.70	−15.08	−10.39	−3.22	uncharacterized LOC115262616
LOC115266688	GZ	−7.77	−43.50	−1.39	−12.41	heterogeneous nuclear ribonucleoprotein K-like
LOC115267355	GZ	−1.90	−7.99	−3.94	−26.05	alkyldihydroxyacetonephosphate synthase-like
LOC115269716	GZ	−1.32	−35.84	−9.90	−∞	Peritrophin-1-like
LOC115270118	GZ	−1.42	−6.09	−3.17	−4.60	Sialin-like
LOC109405426	JH	−1.26	−6.14	−1.62	−5.43	probable cytochrome P450304a1
	GZ	−1.23	−8.95	−2.52	−24.22	

## Data Availability

The clean reads of this study have been deposited into the CNGB Sequence Archive (CNSA) of the China National GeneBank DataBase (CNGBdb) with accession number CNP0003800.

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
