# Peer review of "Transcriptome Analysis of Response to Zika Virus Infection in Two Aedes albopictus Strains with Different Vector Competence"

_ijms, 2023, doi:10.3390/ijms24054257_

Round 1
Reviewer 1 Report
I would like to extend my sincere gratitude to the authors for giving me the opportunity to review their paper. it was a privilege to be able to contribute to the review process and provide my insights and recommendations of their work. I appreciate the time and effort the authors have put into their research, and it was a valuable experience for me to be able to review their findings. I am confident that the feedback I provided will help the authors improve their work and contribute to the advancement of knowledge in this field. Thank you again for allowing me to be a part of this process.
Here are some comments I would like to suggest to the authors.
Introduction: good
Results:
Figure 3, 4 and 5. It is very difficult to read and it does not provide my data to analyze- move to supplement is possible.
Figure 7 and method 4.11: Provide more detail of the sample size injected. Number of dead post siRNA and Post ZIKV injection. Also, if possible, the authors should provide the siRNA knockdown efficiency.
Methods: good
Discussion: Provide more detail of the 304a1 or CYP304a1 gene regarding its role in immunity in other insect if possible.
Final thought:
This is a well written paper and well thought out study. One other thing I would like to suggest is the do a tissue specific expression of CYP304a1 analysis and then follow by post blood meal time course analysis. These results might shed light on the role of CYP304a.
Author Response
Please see the attachment, thank you.

Reviewer 2 Report
As an entomologist I could make some observations concerning my scientific beliefs. The fact that the vector, Aedes albopictus, can be the primary vector of zika in one territory and not in another is linked to the virus, the reservoir of the virus, and the vector (Ae. albopictus), i.e. to the vector competence studied in this work. In certain ecological contexts extrinsic to the mosquito and biological factors intrinsic to the mosquito, a VBD occurs.
Among extrinsic factors, to be remembered, as the territory of one province or another, movements of persons, globalisation of goods and people. But in this work, information on the origin of the mosquitoes(Wild or lab reared) used and the representativeness of the samples was not given information.
However, same authors stating if for Ae. aegypti the biological mechanisms upon occurrence of Zika are known, same for Ae. albopictus there is still no explanation . I confess that my molecular knowledge is not high enough to evaluate the single lab studies , but they run a valuable job . Getting infected by a Virus then multiplication in mosquitoes, eventually virus transmission to a new reservoir (humans...) is not the same in different strains of one species, different strains of a mosquito species seems as if they are different species.
Author Response
Please see the attachment, thank you.

Reviewer 3 Report
The recent re-emergence of Zika virus and it’s impact in Southern America, especially Brazil and the resulting cases of microcephaly has brought this virus back into focus. Since 2015, the molecular and genetic relationship between Zika and its primary carrier, Aedes aegypti, has been extensively studied. However, such studies are limited in other Aedes species. Aedes albopictus is prevalent in temperate climate zones and can facilitate a wider potential outbreak of this disease if equally efficient. Authors Jia et al. compared the vector competence of two Ae. albopictus strains from the Jinghong (JK) and Guangzhou (GZ) regions of China, with particular focus on the two primary virus infection and replication barriers- midgut and salivary glands. Overall, JK strain was found to be relatively more resistant and GZ more susceptible to Zika infection, potentially raising concerns about spreading the virus in temperate zones.
Comments-
1. It is interesting to note that JH strain midgut shows resistance to viral infection initially but then increases to a comparable level to GZ strain at 14 dpi. A detailed possible explanation/cause of this should be provided and further studies performed, as midgut is often the primary barrier to a successful viral infection. From Fig 1A and 1B, it is clear that a barrier exists in the JH strain till 10dpi, this needs to be explored further.
2. Nearly 10 fold upregu;ated DEG in salivary gland of GZ strain
3. Find the dpi which leads to breakdown of barrier in JH strain i.e a point between 10 and 14 day. Repeat mRNA lib seq from this day point and seek common highly upregulated or downregulated genes.
4. The RNAi experiment knocking down CYP304a1 and intrathoraric injection of a high virus titre is not well designed. As authors themselves suggested, the route of entry could potentially bypass the regular route, thus not impacting midgut escape barrier. This study should be repeated with a) the low and high doses used in infection experiments b) through membrane feeder method so as to be consistent with the other experiments.
5. The study also excludes any DEGs which did not have same direction of regulation in both the tissues, thus leaving out potentially relevant genes from the analysis.
Minor comments-
1. Pg 237-238- ..”so it should not be ignored that the risk of ZIKV transmission by Ae. albopictus in Jinghong city.” Does not make grammatical sense. Please correct it.
Author Response
Please see the attachment,thank you.

Reviewer 4 Report
1. The authors aimed to analyze the difference in the vector competence of the two mosquito strains, which may be an important issue, but the key factor CYP304a1 was screened out based on concurrent down-regulation, not the opposite regulation in the two strains. The authors should reconsider the strategy or further give a detailed discussion.
Also, the function analysis by siRNA CYP304a1 knock-down showed no significant effect on virus infection or replication.
This weakens the novelty of this paper and makes the current supposed main idea not suitable.
2. Line120: the authors chose (1×107 120 PFU/mL) on 10 dpi as the condition for transcriptome analysis, they should give an explanation.
3. In section 2.6 why was thoracic microinjection chosen for viral infection instead of the oral infection of mosquitoes? Especially since CYP304a1 was screened out based on oral infection, not microinjection.
4. What is the concentration of viral infection in section 2.6? Considering the difference between low dose and high dose of ZIKV in Figure 1, the dose might have a significant effect on results.
Author Response
Please see the attachment,thank you.

Round 2
Reviewer 3 Report
I am satisfied with the authors' responses to my previous comments/suggestions. Re-performing a better designed mRNA seq study would be a way forward to further explore the factors involved in viral resistance in the JH strain.
Reviewer 4 Report
The authors have addressed the primary concerns of the previous review report in the revised paper and reply. It could be accepted for publication.